# Entrepreneurial Potential and Agribusiness Desirability among Youths in South Kivu, Democratic Republic of the Congo

Guy Simbeko [1,2,*], Paul-Martin Dontsop Nguezet [3], Haruna Sekabira [4], Mastewal Yami [5], Serge Amato Masirika [6], Krishan Bheenick [7], Deogratias Bugandwa [2], Dieu-Merci Akonkwa Nyamuhirwa [2,8], Jacob Mignouna [6], Zoumana Bamba [8] and Victor Manyong [9]

1   International Institute of Tropical Agriculture (IITA), Agricultural Transformation Agenda and African Agricultural Leadership Institute (ATA/AALI–DRC), Bukavu P.O. Box 062, Democratic Republic of the Congo
2   Faculty of Economics and Management, Université Catholique de Bukavu (UCB), Bukavu P.O. Box 285, Democratic Republic of the Congo
3   International Institute of Tropical Agriculture (IITA), Kalemie P.O. Box 570, Democratic Republic of the Congo
4   International Institute of Tropical Agriculture (IITA), Kigali KG 563 #3, Kigali P.O. Box 1269, Rwanda
5   Independent Researcher, Addis Abeba P.O. Box 5689, Ethiopia
6   International Institute of Tropical Agriculture (IITA), President Olusegun Obasanjo Research Campus, Kalambo, Bukavu P.O. Box 1222, Democratic Republic of the Congo
7   Capacity Development Consultant, Food Organization of the United Nations (FAO), Accra Isert Road, No. 69, North Ridge, Accra P.O. Box GP 1628, Ghana
8   International Institute of Tropical Agriculture (IITA), Kinshasa 4163, Democratic Republic of the Congo
9   International Institute of Tropical Agriculture (IITA), Daar Es Salaam 34441, Tanzania
*   Correspondence: simbeko.sadi@ucbukavu.ac.cd or g.simbeko@cgiar.org; Tel.: +243-993429730

**Abstract:** In the Democratic Republic of Congo (DRC), entrepreneurship in the agriculture sector remains for youth a key pillar for income creation. However, few are attracted by agribusiness despite stakeholders' efforts toward engaging youth in agriculture. Therefore, this study examines the relationship between entrepreneurial potential characteristics and youth desirability to start an enterprise in agriculture among 514 young people in Eastern DRC. This study revealed that youth in South Kivu have different entrepreneurship potential features and agribusiness desirability levels according to their gender and living area. Hence, the youth's agribusiness desirability is motivated by an awareness of emerging agripreneurial activities, land ownership, parent involvement in farming activities as a role model, perceived agribusiness as an employment source, management-organizing and opportunistic competencies, market analysis, negotiating, and planning skills. Therefore, efforts to attract youth into agribusiness should focus on the use of media, the creation of awareness of available agribusiness initiatives in their area, and the setup of land policy. This is in addition to putting in place capacity-building programs on entrepreneurial and business skills through incubators, and the formalization of youth agribusiness groups that foster capitalizing experiences between new and accelerated agripreneurial enterprises, with the support of parents and financial institutions, focusing on gender sensitivity, in both rural and urban areas.

**Keywords:** entrepreneurial potential; agribusiness desirability; agribusiness; youths; DRC

## 1. Introduction

The notion of entrepreneurial resilience stipulates that a successful enterprise requires potential entrepreneurs who can turn innovative ideas into attractive business opportunities [1–3]. Entrepreneurial potential characteristics are needed for this human capital to serve the community by reducing unemployment, and ultimately, increasing people's income and resource efficiency for sustainable development [4,5]. Nowadays, agricultural economics policies and practitioners of development pay special attention to entrepreneurship as a promoter of pecuniary activities to ensure financial autonomy and enhance

livelihoods and people's well-being mostly in rural areas, which is a vital element for the economic development of nations at the world scale [6–8].

Hence, agriculture is conceived as a key part of the economy as more than 60% of the world's population depends on it for survival [9]. Literally, in Africa, 70% of its population depends on farming activities within the majority of its people are below 30 years old. The continent has the youngest population in the world and mostly lives in rural areas with restrained employment possibilities [10]. In such a context, entrepreneurship is one of the most desirable ways for decreasing youth unemployment [11,12]. However, the trend is somewhat varied. There is a relatively slow progression rate by the youth to take advantage of entrepreneurship and self-employment, especially in the agricultural sector [13,14]. Using Demographic and Health Surveys (DHS) data from 19 African countries in 2013, [15] estimated the probability of working in agriculture for people aged between 25 and 60 years during the 2000–2010 period, and the results have shown a 10% decline in the share of labor engaged in agriculture. This decline corresponds to an 8% increase in the share of labor in services and a 2% increase in the share of labor in manufacturing during the same period [13], and, as of now, the trend is still mitigating considerably.

In the South Kivu province, the Eastern part of DR Congo, the unemployment rate of youths who have completed their undergraduate studies is critical. On average, only one graduate out of a hundred is annually employed [16] and the situation is getting steadily worse since the COVID-19 pandemic, coupled with demographic pressure that affected many employment opportunities. However, programs are intervening in the province to reduce youth graduate unemployment through entrepreneurial development potential features as it is important for participants in the modern agriculture and food industry for sustainable agribusiness. The most principal programs are IYA and IITA-Youth in Agribusiness, which seeks to promote graduated youth involvement in agribusiness. The Young Professionals for Agricultural Development programs supported by the Forum for Agricultural Research in Africa (YPARD-FARA) accompanies youth in their professional integration under the youth entrepreneurship in Agriculture, and the Great Lakes Integrated Agriculture Development Project [17] aims to increase agricultural productivity and marketing in the province, and also improve regional integration in the agricultural sector within youth engagement in agribusiness.

Studies and projects in the last decade in South Kivu have linked psychological traits and discriminating factors within sexo-specific parameters to explain youth intention to start a venture in agriculture [16,18–23]. However, few of them had focused on entrepreneurial potential features that need to be involved in agribusiness activities. Besides, studies that have attempted to link entrepreneurial potential issues and agribusiness have mostly been either qualitative, focused on in-depth case study approach and coupled with a review or narrative textual cases [24–28], or conceptual. Other studies conducted on the determinant of agriculture entrepreneurship have just taken into account cognitive factors, social capital, and demographic factors, ignoring other entrepreneurial potential parameters [16,29]. Moreover, various types of research in Africa on youth engagement in agribusiness [30–35] have demonstrated that the Theory of Planned Behavior (TPB) and the sustainable livelihoods framework (SLF) can be used to predict youth intentions in agribusiness. According to the theory of planned behavior (TPB), behavioral intentions, the immediate precursors of behavior, are determined by the subjective norm for behavior and perceived control over behavior [36]. Furthermore, most of the shortcomings of livelihood research and assessment like sustainable livelihoods framework (SLF) are due to the fact that the use of such a framework has been only theoretical [37]. Nevertheless, personality traits and cognitive ability are hidden competencies as argued by [36]. To address the problem with these two kinds of theories, researchers suggest by first understanding what people can do to earn a living, and why they have made these choices as a way to understand how their choices and strategies have been shaped. Researchers have assumed that behavior generally serves as a means to an individual's end. To expand the scope of TPB and increase its explanatory and predictive power, they integrated into a proposed Theory

of Reasoned Pursuit of Goals (TRGP). Hence, this study opted for the Theory of Reasoned Pursuit of Goals (TRGP) developed by [37], in such a way that agriculture entrepreneurship is a decision of individuals to start a new agriculture business [38]. In addition, the desire to engage in a farming activity depends primarily on the perceived likelihood that the application of entrepreneurial potential skills in the farming activity will lead to achievement and success in agri-food development goals attainment. This study hypothesizes that entrepreneurial potential coupled with socio-economic factors could positively affect the desirability level of South Kivu youth to start a venture in agriculture as a further prosperous career. This implies incorporating measures of active entrepreneurial skills (communicative, strategic, organizing, opportunistic, and social demographic factors) as a factor that can drive attitudes, motivation, intention, and subsequent desire to be involved in agribusiness. As the main purpose, this study addresses empirically the relation between entrepreneurial potential factors and Agribusiness Desirability (AD) in the context of the South Kivu province.

To go beyond the TPB and the SLF and then contribute to the literature on agribusiness desirability in South Kivu, this study aims specifically at: (1) assessing the entrepreneurial potential and agribusiness desirability level of youth in the South Kivu province in the towns of Bukavu and Uvira and the territories of Kabare and Walungu, using Exploratory Factor Analysis (EFA); (2) to determine the effect of potential entrepreneurial competencies variables on youth agribusiness desirability level using structural equation approach; and (3) to bring out within entrepreneurial competencies the effect on agriculture entrepreneurship of control variables, namely age, gender, marital status, education, farmers membership in an association, income, parental support, land access, parental involvement in farm activities as a role model, awareness about agriculture initiatives emerging in the community, enrollment in agricultural training, and the perception of agriculture as a source of employment, by using truncated regression. Therefore, this study first contributes to the existing knowledge body on the Theory of Reasoned Pursuit of Goals (TRGP) by showing that there is a significant correlation between the desire to engage in agribusiness as a reasoned goal pursued by youth through applying potential entrepreneurial skills coupled with the significant role played by youth's socioeconomic and demographic factors in agribusiness. Secondly, this study fulfills the gap that needs to be elucidated manifesting in the research by [24]; [9,38,39] by tackling the promotion and development of agricultural entrepreneurship through a holistic approach among youth in contexts of developing countries through exanimation of the link between EP and AD. This approach is radically new to this sector and consistent with the seminal work of [40,41], which included control variables, demographic and economic, and entrepreneurship components in their study on agriculture entrepreneurship. Our findings will inform young people and their communities, but also policymakers and other stakeholders whose interventions seek to involve youth in agribusiness and rural development with agriculture entrepreneurship, and specifically the options on which to build on, in order to increase the level of youth participation in agribusiness. In this study, a youth is defined as any individual between the ages of 15 and 35, according to the African Union Charter [42]. However, we start consider in sample youth between the ages of 18 and 35, which, according to [2], is the adult age when a youth individual is able to make a rational choice and is responsible for his or her orientations. Apart from the introduction, the next section of this paper presents a literature review. Then, the materials and the methods used. Finally, the results and the discussion are presented in the third section, and limitations and recommendations are presented in the very last section.

### 1.1. Entrepreneurial Potential (EP)

The entrepreneurial potential is defined by [9,43] as the extent to which an individual possesses the characteristics that are associated with successful entrepreneurs. Focusing on the manifestation and measurement, [43] assumed theoretically that entrepreneurial potential is a latent expression of the perceived desirability, perceived feasibility, and propensity

to act. Other empirical approaches, such as [44], suggest that entrepreneurial potential is expressed by seven characteristics: the need for achievement, locus of control, propensity to take risks, problem-solving, willingness to assert oneself, tolerance of ambiguity, and emotional stability. Based on [36] the Theory of Reasoned Pursuit of Goals (TRGP), motivation to perform a behavior depends to a large extent on active goals, not inactive goals. Hence, the intention to adopt a behavior is the facility of the ability to implement it [5,6], and, therefore, the intention to do agribusiness in this study is also a function of ability from a rational choice made up by a youth base on entrepreneurial and socio-economic competencies possessed. Ahmad [38] report the entrepreneurial competencies that are required for the viability of a business include 12 main clusters, namely, strategic, commitment, cognitive, opportunity-based, organizing and leading, communicative, learning, personal, technical, ethical, social responsibility, and support and cooperation competencies. The authors of [41,45] discuss the competency requirements of an entrepreneur in a framework composed of behavioral, communicative, and foreign relations competencies. By taking into account multiple entrepreneurial competencies and the research on them, Ref. [6] categorized these competencies into five traits: psychological, strategic, organizing, communicative, and opportunistic. The later demonstrated that entrepreneurship is not exclusively the result of an individual's actions and characteristics at an individual stage, and external factors such as the economic, technological, political, and regulatory context also plays a relevant role [38,46] at country level. Hence, this paper focuses on the individual, and also considers that environment factors could made up entrepreneurial potential competencies [45]. This work considers that entrepreneurial potential refers to the sum of several individual entrepreneurial skills acquired either from an innate way or from an environment context which predisposes someone to engage in entrepreneurial activities. Thus, this study aims to highlight the entrepreneurial skills that are most associated with starting a business such as agriculture. Furthermore, individuals, particularly youth, are the main agents who assume all consequences in the process of involvement in agriculture entrepreneurship activities and then decide to appeal to his/her potential to succeed in the goal pursuit in his or her environment. Inspired by the studies of [6,24,47], this study used five entrepreneurial potential features; psychological, strategic, organizing, communicative, and opportunistic characteristics required for the viability and agripreneurship performance in developing countries, mostly in a rural context. The conceptual model (Figure 1).

### 1.2. Agribusiness Desirability (AD)

Made up by [8] and cleared up by [48], agribusiness is the business of agricultural production which involves the production, protection, sales, and marketing of the product to satisfy customers' needs. The term is a pillar of agriculture and business and it includes agrichemicals or genetics and seed stocks, breeding, crop production, all agents of the food and fiber value chain distribution, farm machine, processing, digital agriculture, seed supply, as well as marketing and retail sales to consumers. In accordance with [36], the Theory of Reasoned Pursuit of Goals (TRGP), and the work of [49], this study defines perceived desirability as the personal attractiveness of starting a business, including both intrapersonal potential and extra active personal impacts beyond other life expectancy objectives. Agripreneurship involves taking risks and accepting uncertainties to develop a business venture to create a profit or return on an investment [48,49]. Thus, any individual, therefore, who develops innovative ways to invent, transform, or create a product or service within the agricultural value chain, including value addition to existing products, while bearing the risks, would be considered as an agripreneur [48]. Thus, the AD as a career choice among youths will increase with the perception that such activities are supported within their networks. This paper applied eight (8) items inspired by the studies [9,50–52], and adapted them to the context of South Kivu to capture youth agribusiness desirability levels.

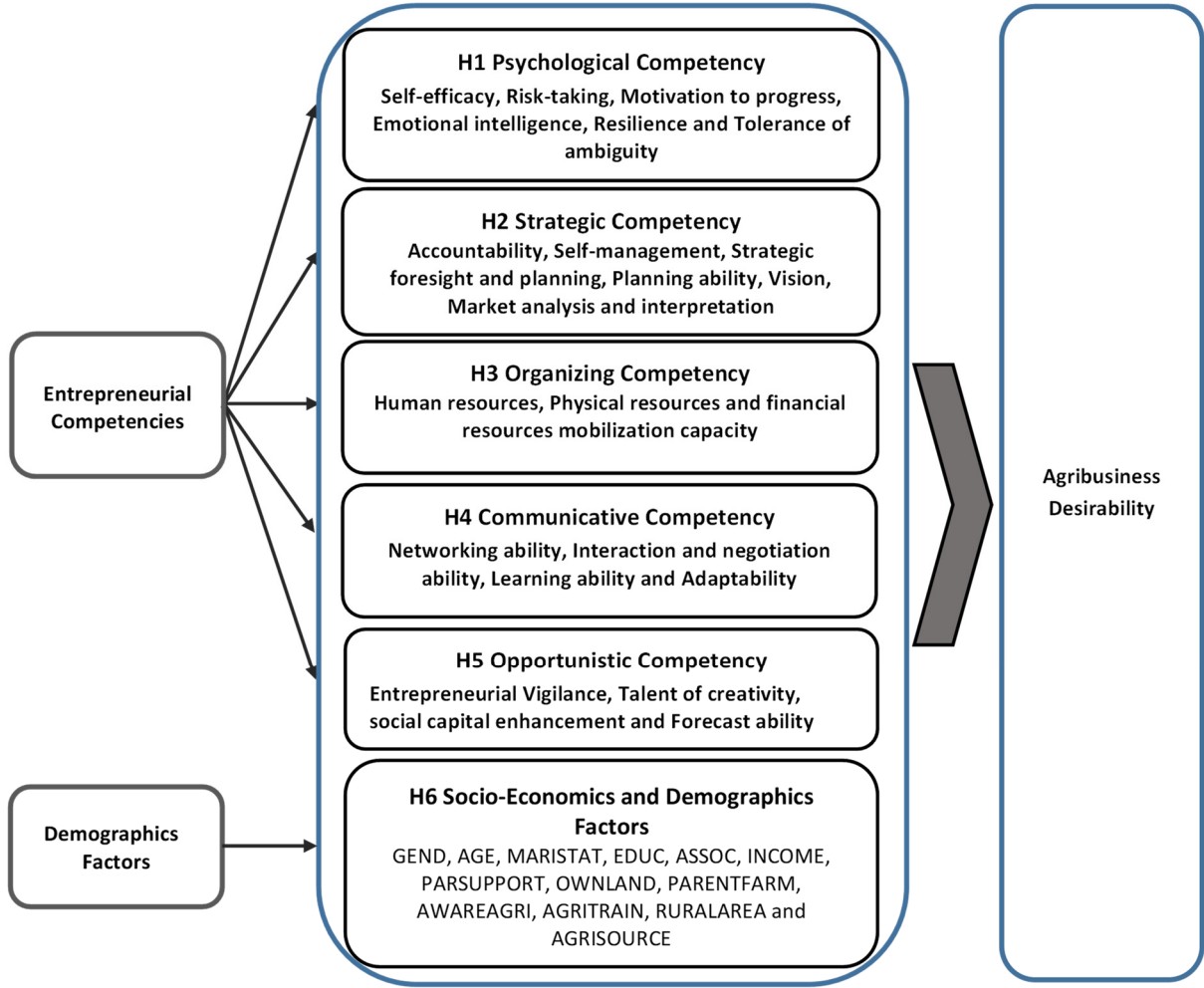

**Figure 1.** The author's conceptual model of entrepreneurial potential and its effect on youth's intention to launch SMEs in rural contexts is inspired by [6,24,40], and is adapted to agriculture entrepreneurship desirability. Note: GEND: gender; ASSOC: membership in community association/Group; EDUC: education level; INCOME: monthly income from a different source of funding; PARSUPORT: rate of parental support in income; OWNLAND: land access; PARENTFARM: parents involved in the farming activity (as a role model); AWAREAGRI: aware of agriculture initiatives emerging in the community; and AGRISOURCE: youth perceived agriculture as a source of employment.

*1.3. Theoretical Framework and Literature Review between EP-AD*

Entrepreneurship is a process of taking a risk by an individual or group of people for creating profit and new values throughout input (work, capital, and land) combination [6]. Furthermore, [53,54] define intention as a person's disposition to perform a given behavior such as the individual's desire to pursue a career as an independent. Typically, an individual identifies a need or problem within a community, and afterwards becomes a social entrepreneur by starting and developing a social enterprise [51]. All these things considered, the main motivation for any commercial enterprise is usually to obtain profit; however, the ultimate goal is social impact and plays an important role to bring change within the social sphere. They also assume a mission of creating and perpetuating social as well as economic values [1]. Hence, entrepreneurship oriented in the agricultural sector becomes congruent to fight against poverty, generate rural employment, reduce migration from rural to urban areas [2], with regards to socio-economic welfare and sustainable agriculture value chain development [3,4]. Furthermore, in agriculture entrepreneurship studies, factors were also studied along with demographic factors to understand and profile entrepreneurs better. Hence, agriculture entrepreneurship is not innate; most of the

entrepreneurial potential that causes agripreneurs to succeed is acquired through learning, formal, and informal manifest entrepreneurial competencies. Hence, researchers [19,36,54] assume that entrepreneurial behavior is planned, reasoned, and controlled in anticipation of likely consequences. The intention to launch a business depends upon on the individual's psychological capital [55,56] and entrepreneurial skills [6,33]. The essential ingredients of psychological capital include self-efficacy, optimism, hope, and resilience [57]. Entrepreneurial skills are communication, planning, networking, creativity and the ability to identify opportunities in one's environment [6,58]. Resilience is the extent to which individuals are able to bounce back or recover from negative experiences, failures, and adapt to changing and stressful life circumstances [59]. In order to do this, critical skills must be developed, including creativity, flexibility, and adaptability. Hence, research by [16] supports a positive relationship between psychological capital and agripreneurial intention. A positive relationship between psychological capital and agripreneurship can be justified by the fact that self-efficacious youth will believe in their abilities to succeed in achieving entrepreneurial behavior [9,25]. In addition, optimistic youth can recognize business opportunities where others see chaos, contradiction, and confusion [32]. Furthermore, the dimension of hope helps them capitalize on these opportunities by setting high goals that they believe can be achieved because they can see a way to success and resilient youth can take risks and bounce back from setbacks and adversity [19,60]. Other researchers have shown that self-efficacy refers to beliefs in one's abilities [61] or the level to which a person feels able to mobilize the motivation, cognitive resources, and courses of action necessary to successfully complete a specific task. By using a multivariate analysis approach on assessing farmers' entrepreneurship skills in agriculture and the competitiveness of the small and medium enterprises, [5,6], in Iran, and [38,54], in Nigeria, categorized five entrepreneurial potential skills which included networking ability, individualism, tolerance of ambiguity, and market analysis, as a forecast of rural youth's intention to launch any kind of SMEs. On assessing the effect of norms perceived on agripreneurial intention among youth in eastern DRC, [13] indicated that psychological capital has positively and significantly affected youths' agriculture entrepreneurship intention. For [11,14], potential entrepreneurs in agriculture with higher psychological competency have high risks and potential drawbacks. Other researchers [2,16,17] demonstrated that entrepreneurial and organizational competencies have a positive effect on sustainable development agriculture among farmers. The managerial, technical, and innovative skills of entrepreneurship applied to agriculture have positive results on their yield and well-trained entrepreneurs may become role models to all such disheartened farmers [16]. According to [16], management skills are the complete package of skills that an entrepreneur in the agriculture sector would use to develop the farm business. That means managers perform various agricultural-based activities to mobilize different resources—physical, financial, human resources, and information—to accomplish the agriculture entrepreneurship purpose. The authors of [6,40] demonstrated that organizational competencies affect positively enterprise lunching in rural area. For [11], the ability to interact effectively with others has a positive effect on entrepreneurial success. The ability to develop a network between entrepreneurs and other individuals that provide them with resources to create and develop the enterprise has been identified as one of the predictors of entrepreneurial performance [17]. The authors of [12,18] attested that in the agriculture sector, individuals who have entrepreneurs in their networks may also have access to the resources important for starting a business [19]. These resources may include technical knowledge; contact with business class; and emotional support from their community, including friends and family members. All put together, high psychological capital and entrepreneurial skills in agriculture entrepreneurial intention could allow young people to exhibit high agripreneurial intention. Following the different links presented in framework and literature between agricultural entrepreneurship and the different entrepreneurial competencies, this study offers the following hypotheses.

**Hypothesis 1 (H1).** *Psychological competency (PSYCOMP) has a positive effect on youth's agribusiness desirability (AD) among youth in the eastern part of DR. Congo.*

**Hypothesis 2 (H2).** *Strategic competency (STRCOMP) has a positive effect on youth agribusiness desirability among youth in the eastern part of DR. Congo.*

**Hypothesis 3 (H3).** *Organizing competency (ORGCOMP) has a positive effect on youth's agribusiness desirability among youth in the eastern part of DR. Congo.*

**Hypothesis 4 (H4).** *Communicative competence (COMCOMP) has a positive effect on youth's agribusiness desirability among youth in the eastern part of DR. Congo.*

**Hypothesis 5 (H5).** *Opportunistic competency (OPPCOMP) has a positive effect on youth's agribusiness desirability among youth in the eastern part of DR. Congo.*

*1.4. Socio-Economics and Demographics Factors and AD*

Following many previous studies, this study also took into account demographic variables in the present research. There are several pieces of evidence which suggest that to control variables influencing agriculture entrepreneurship and entrepreneurial behavior and sustaining the enterprise, demographic variables are important. In this study, the majority of the respondents were literate and living in rural areas. Hence, literacy is expected to influence their perceptions of information received and utilized for agricultural activities. Furthermore, educated people are expected to accept a moderate degree of awareness about agricultural activities. Research by [52] noted that youth have a greater knowledge acquisition propensity; therefore, they are eager to discover new ideas or inventions. Some scholars [62,63] related that education has a positive influence on intentions, attitudes, and self-efficacy. Other scholars, however, have observed a negative relationship between education and agripreneurship intention [16,55]. The kind of link between having a higher education level and living in a rural area within the development of agriculture entrepreneurship remains mitigated and the trend is changing among youth according to their living area. Hence, this study aims to investigate the effect of education on agribusiness desirability in a rural and urban context.

A study conducted by [21] in South Kivu shows that females who were not involved in farming had a higher percentage of people not interested in agriculture compared to males. Therefore, more females who were involved in farming had a better interest and perception of agriculture compared to males. Similarly, women are disproportionately under-represented in most formal sectors of the economy, except in agriculture, which coincidentally remains predominantly subsistence-based and underdeveloped. This study posits that gender could have a positive effect on Agribusiness Desirability (AD) among youth in the South Kivu province. In addition, access to land is also cited among the factors influencing youths' intentions to engage in agribusiness. Access to land is a prerequisite for most types of agribusinesses [64,65]. Land ownership is an essential asset for agriculture. This study posits that agribusiness desirability is strong among youth-owned cultivation portions of land.

Furthermore, belonging to a group or association can significantly improve the usefulness of young people involved in agribusiness. Association membership allows entrepreneurs to have the bargaining power to discuss market shares and reduce transaction costs. Further, membership in a group expands its network and therefore allows relatively easy access to information [56]. Therefore, being a member of a group would be positively linked with the chosen character of investing in the agroindustry. Knowing agriculture initiatives emerging in the community is among cited in the literature [57] as an element which emphasizes the agripreneurship intention among youth. In [58], a study using a binary model to analyze the determinants of youth participation in agriculture in the Nkonkobe Municipality in South Africa, results have shown that variables such as the youth entrepreneurial program and resource availability were statistically significant in

explaining the factors that affect youth participation in agricultural activities. Accordingly, this study posits that awareness about agribusiness activity could have a positive effect on youth's agribusiness desirability in the context of South Kivu. Relatively, a study [59] on the impact of marital status on youth entrepreneurship in Malaysia has shown that subjective norms do not affect unmarried people, but married people are affected by subjective norms. In [23], the same evidence among youth entrepreneurs was observed in South Kivu and we expected this could be the case in this study.

In addition, providing training to employees to develop their knowledge and skills can improve the quality of human resources [60]. This study presumes that youth living in an area where many entrepreneurial-capacity buildings are involved could shape their perception of agriculture as an employment source, could increase their awareness of agricultural initiatives emerging in their community, and could provide the possibility for them to be trained or to receive exposure to entrepreneurial skills within an agricultural entrepreneurship background. In addition, several studies have highlighted the importance of income within the entrepreneurial process. A study by [61] demonstrates that a high income increases the likelihood of starting a business because it offers the possibility of freely accessing the information sought and overcoming the financial barriers associated with setting up a business. Furthermore, [66] explain that high income is an asset that influences job creation and reduces financial constraints to business creation. Considering the above evidence of income in the entrepreneurial process, this study assumes a positive relationship between youth income and agriculture entrepreneurship. Equally, perceptions play an important role in influencing the interests of the youth in agripreneurship. According to [46], individuals decide to start an entrepreneurial activity if it is perceived to be more desirable and feasible than other alternatives. Other studies have found that individuals whose parents are entrepreneurs often become entrepreneurs because of their perceptions that might be formed through observing these role models [62] mostly in the context of South Kivu, where rural youth face barriers that are related to the imperfect markets, and incomplete information often characterizes the entrepreneurial environment, and others are occasioned by socioeconomic bottlenecks. Particularly in cases where career guidance is limited or not easily accessible, an individual's views or perceptions may become the most influential factor in making a decision. Hence, this study proposes that Agribusiness Desirability (AD) could be increased among youth whose parents are involved in farming activities. Hence, following socio-economic and demographic parameters from the literature, finally, this paper hypothesized the following.

**Hypothesis 6 (H6).** *Demographic parameters within socio-economics and demographic variables; education level, gender, age, marital status, membership in the association, parental income support, land ownership, parental involvement in farming activities, awareness about agriculture initiatives, enrollment in agriculture training, living in a rural area, and perceived agriculture as a source of youth employment have a positive effect on youth's AD among youth in South Kivu.*

## 2. Methodology

The data used covered 514 students randomly selected from 20 universities throughout urban and rural areas of the South Kivu province including the Bukavu, Uvira, Walungu, and Kabare territories, from October to November 2020. The choice of these areas (Figure 2) is justified by their high levels of youth unemployment, prominence of smallholder farms, and a large number of private initiatives, government, and NGO programs, which intervene to reduce youth unemployment. With academic staff, including a professor and lecturer, in addition to student representative support, we have identified and established a list of 1750 students without any venture in agriculture from which we have applied a random selection and obtained 514 respondents aged between 18 and 35 years. Primary data were collected through smartphones equipped with the Open Data Kit (ODK) collect software. Respondents were found at their campuses, homes, or workplaces.

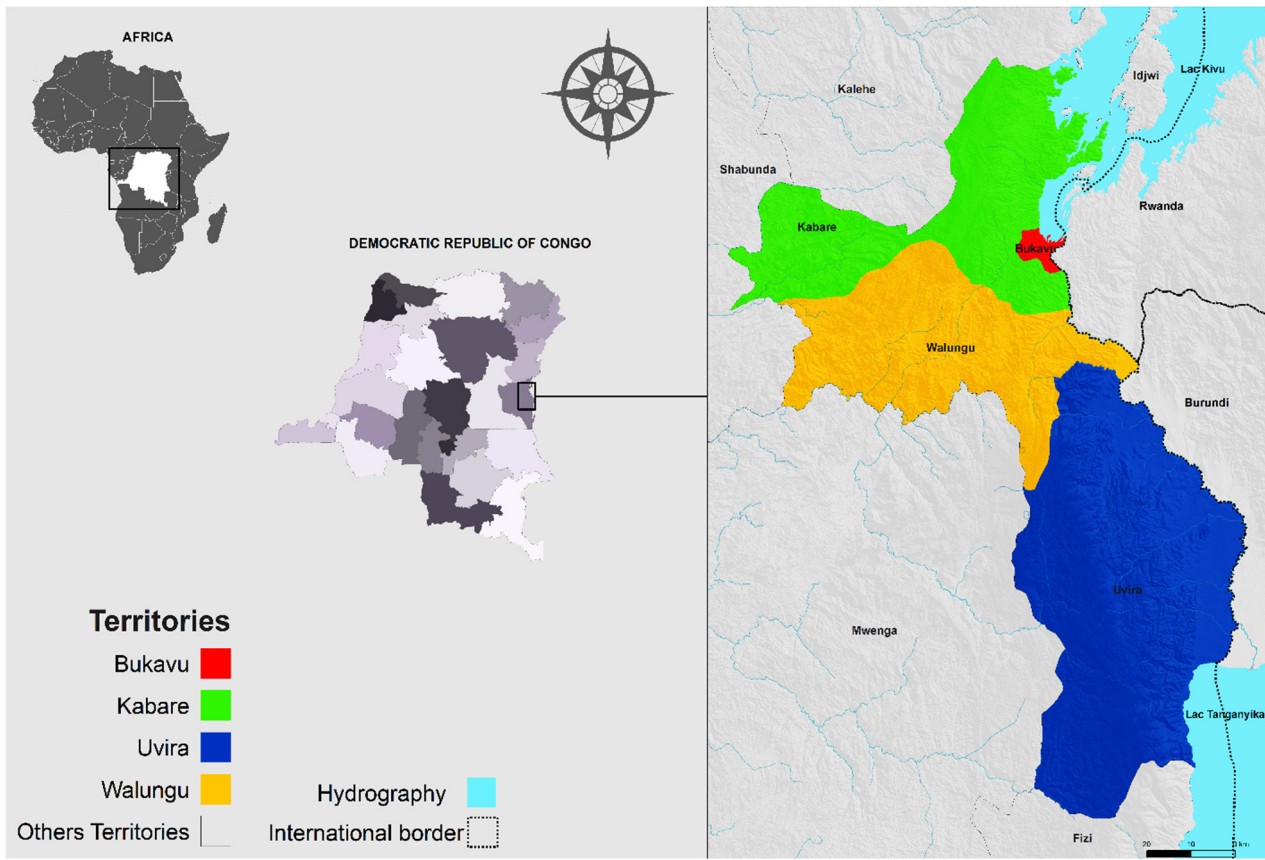

**Figure 2.** Map of the research area. The left map shows the position of the South Kivu region within the territories and towns surveyed. The right map shows the position of the province wards within the D.R Congo's eastern part of Africa.

Exploratory Factor Analysis (EFA) was used to discover the factor structure of a measure and to examine its internal reliability. In addition, to improve the model quality, we removed any item whose loading coefficient was lower than 0.5 in the explanation of the construct. To collect data, a questionnaire was used that faced a panel of experts. The pairwise comparison confirmed content validity, and a pilot study confirmed its reliability. In the first part, the questionnaire captured socio-economics and demographic variables: age, gender; membership in a community association, education level, monthly income, rate of parental support in income, land access, parental involvement in farming activity (as a role model), aware of agriculture initiatives emerging in their community, and youth agriculture perception as a source of employment. In the second part of the questionnaire, we took into account strategic, communicative, psychological, and opportunistic competencies, which were the most to display important factors from the literature underpinning the rural youth's entrepreneurial intention to establish SMEs in developing countries, and specifically, in rural areas. The variables used in this study were measured using multi-item measurement scales inspired by previous empirical studies in agriculture entrepreneurship [6,19,24,47]. In addition, the advantage of this type of scale is that it minimizes measurement errors, thus maximizing the validity and reliability of the questionnaire. For the endogenous variable and the exogenous variables, a 5-point Likert scale ranging from "1": strongly disagree to "5": strongly agree was used to collect the opinions of youth. SPSS 20.0 and SmartPLS 3.3.3 software were used respectively for data epuration and analysis. The dependent variable Agribusiness Desirability (AD) was measured using 12 items inspired by studies by [16,52,67,68]. The psychological skills variable was measured through 14 items inspired by the literature [63,69] and grouped into 4 dimensions including individualism, risk-taking, motivation to progress, self-efficacy,

and tolerance of ambiguity. A total of 10 items divided into 4 dimensions measured the strategic competencies variable: accountability, self-management, planning ability, market analysis, and interpretation. Organizing competencies were measured by 14 items grouped into 3 dimensions (human resources, physical resources, and financial resources). Communicative competence was measured by four dimensions (networking ability, interaction and negotiation ability, learning ability, and adaptability) divided into 11 items recognized by the literature [67]. Opportunistic competency was measured by 7 items inspired by [68] and grouped into 3 dimensions, namely opportunity recognition, creative talent, and social capital development. The measurement scales were slightly adapted to the context of our study environment. Thus, the reliability and validity of the measurement scales were tested. The variables were introduced as unidimensional into the model. This allowed us to directly observe the effect of each skill on the desirability of agribusiness.

*Reliability and Validity of the Measurement Scales*

Before factor analysis, we computed the Kaiser Meyer Ohlkin (KMO) index and tested it with the Bartlett test to confirm that the items were suitable for factor analysis. The dimensions to be retained had to exhibit an eigenvalue > 1, which meant that they were better than one item. Any item that contributed to reducing the overall reliability of the measurement scale was removed. This was assessed by Cronbach's alpha coefficient, which had to be greater than or equal to 0.7. We also deleted any item whose quality of representation (communality) in the measurement scale was lower than 0.5. In addition, to increase the convergent and discriminant validity of the measurement scale, we deleted any item that did not have a structural coefficient of at least 0.5 on one component or that had at least 0.41 on several components at the same time, following the criteria of [70,71]. Therefore, this study examines the convergence validity of the scales by assessing the composite reliability (CR) of each measure. Mostly, CR values were greater than 0.7, indicating that the reliability and convergence validity of all scales are confirmed according to the criterion of [71]. This study also calculated validity by measuring the average variance explained (AVE). Thus, the AVE values for the scales ranged around 0.5 and above for all measures. Thus, the test of independent variables' effect on the dependent variable was performed using the Partial Least Squares method, using SmartPLS 3.3.3 software (Equation (1)). This method is generally used in the context of structural equation model estimation because the parameter estimators are unbiased, logical, and efficient regardless of whether the assumptions of data normality are satisfied, as suggested by [72]. Further, because the dependent variable is a random variable that takes a value in the interval (1 to 5), we opted for a truncated model in Equation (2) to test the independent variables' (EP and socio-economics and demographics factors) effect on the dependent (AD). The estimation of parameters of this model was estimated using the maximum likelihood method. Inferential statistics were performed to assess the significance of the estimated parameters in the following simultaneous equations model (Equations (1) and (2)):

$$AD = \alpha_1 + c_1 \, EPsi \tag{1}$$

Equation (1) above represents the test of independent variables' (EP) effect on the dependent variable (AD).

$$
\begin{aligned}
AD = \beta_0 &+ \beta_1 GEND + \beta_2 AGE + \beta_3 MARISTAT + \beta_4 EDUC + \beta_5 ASSOC + \\
&\beta_6 INCOME + \beta_7 PARSUPPORT + \beta_8 OWNLAND + \beta_9 PARENTFARM + \\
&\beta_{10} AWARAGRI + \beta_{11} AGRITRAIN + \beta_{12} RURALAREA + \beta_{12} AGROSOURCE + \\
&\beta_1 PSYCOMP + \beta_2 STRATCOMP + \beta_3 ORGCOMP + \beta_4 COMCOMP + \\
&\beta_5 OPPCOMP + e_1
\end{aligned} \tag{2}
$$

## 3. Results

At this point, this work presents the results within respondents' descriptive statistics, the average value of variables used in an empirical model, and results from SEM-PLS analysis 4.1.

### 3.1. Sample Distribution

Most of the young surveyed were in rural areas (39% in Kabare and 29% in Walungu). Out of the 514 interviewed students, 74% were male, and 26% were female (Table 1). The previous could be difficult for women's availability during the survey period given their domestic responsibilities in terms of childcare, housework, and the family in general. According to the International Labor Organization, the gender pay gap is as high as 40 percent in rural areas; rural women take on a disproportionate share of unpaid household work, which is neither recognized nor paid. When paid and unpaid work hours are combined, women work much longer hours than men in the household [6]. This affects their emancipation time for other community development activities.

**Table 1.** Sample distribution.

| Site Surveyed | Area | Female | Male | Total |
|---|---|---|---|---|
| Bukavu | Town | 18 | 42 | 60 (11.7%) |
| Uvira | Town | 33 | 70 | 103 (20.0%) |
| Kabare | Rural | 62 | 139 | 201 (39.1%) |
| Walungu | Rural | 20 | 130 | 150 (29.2%) |
| Total | | 133 (25.9%) | 381 (74.1%) | 514 (100%) |

Source: authors' computation.

The results presented in Table 2 confirm that the data used in this study met the requirements of factor analysis based on the Kaiser Meyer Olhkin (KMO) and Barlett statistics. The KMO statistics of the retained five components were above 0.7 whereas Barlett's sphericity test was significant, meaning that the item correlation was not an identity matrix. The inter-item correlations were not zero. All the selected items had high communalities, showing a strong inter-item correlation.

**Table 2.** Kaiser Meyer Ohlkin test for factor analysis.

| Criteria | PsyComp | StratComp | OrgComp | ComComp | OppComp |
|---|---|---|---|---|---|
| KMO | 0.793 | 0.749 | 0.842 | 0.760 | 0.700 |
| Bartlett sphericity chi-square | 672.19 | 649.64 | 1366.61 | 582.18 | 701.11 |
| Significance (*p*-value) | 0.000 | 0.000 | 0.000 | 0.000 | 0.000 |
| Communality ranges | 0.517–0.729 | 0.537–0.752 | 0.572–0.686 | 0.562–0.641 | 0.601–0.756 |

Source: authors' computation.

In Table 3, the respondents' average age was 25 years, and about 48% have a proportion of land within 0.6, which means it is susceptible to exploitation. A total of 83% of the youth surveyed are aware of agriculture activities carried out in their community. This fact came from many vulgarization initiatives set up by different stakeholders through ICT, such as radio, and social media. The same phenomenon was observed in Uganda and beyond Africa [73]. Most of the youth were unmarried (79%). As evidenced by [66], in Africa, DRC's young women in urban areas are getting married slightly later than in rural areas. Furthermore, 56% of respondents had parents involved in farming activities. According to Table 3, 48% of respondents had access to agriculture training. Thus, 93% of the youth attest that agribusiness is a source of employment for them. This is similar to other empirical studies [20,21,74,75] that make entrepreneurship seem like a panacea for the unemployment problem that plagues the world.

**Table 3.** Descriptive analysis and variables modalities.

| Variable's Description and Modalities | Acronyms | Count (N = 514) | Mean | Std. Dev |
|---|---|---|---|---|
| Gender/Male (dummy %) | GEND | 381 | 74 | 43.8 |
| Years old (Mean year) | AGE | 514 | 25 | 3.57 |
| Married (dummy %) | MARISTAT | 109 | 79 | 40.9 |
| Years of formal education (Mean years) | EDUC | 514 | 16 | 2.08 |
| Membership community association (dummy %) | ASSOC | 216 | 42 | 49.4 |
| Monthly income (Mean USD) | INCOME | 471 | 69 | 64 |
| Parental support financially on youth income (Mean rate %) | PARSUPPORT | 373 | 53.4 | 32.5 |
| Land proportion owned (Mean ha) | OWNLAND | 223 | 0.63 | 0.520 |
| Parents involved in farming activities (dummy %) | PARENTFARM | 286 | 56 | 49.7 |
| Awareness of emerging Agriculture initiatives (dummy %) | AWAREAGRI | 424 | 83 | 38 |
| Enrolment in Agriculture training (dummy %) | AGRITRAIN | 247 | 48 | 50 |
| Living in rural area (dummy %) | RURALAREA | 351 | 68 | 46.6 |
| Agriculture as a source of youth employment (dummy %) | AGRISOURCE | 478 | 93 | 25.5 |
| Psychological competency (Mean) | PSYCOMP | 514 | 3.56 | 0.529 |
| Strategic competency (Mean) | STRCOMP | 514 | 3.27 | 0.531 |
| Organizational competency (Mean) | ORGCOMP | 514 | 3.54 | 0.511 |
| Communicative competence (Mean) | COMCOMP | 514 | 3.57 | 0.609 |
| Opportunistic competency (Mean) | OPPCOMP | 514 | 3.02 | 0.656 |
| Agribusiness desirability (Mean) | AD | 514 | 3.66 | 0.483 |

Note: % is Percentage. Source: authors' computation.

### 3.2. Results from the Exploratory Factor Analysis

The results from the exploratory factor analysis (EFA) in Table 4 revealed that self-confidence and motivation to progress explained about 63% of the Psychological Competencies (PsyComp) variance. EFA on Strategic Competencies shows that StratComp is composed of two components that contribute to explaining 63% of the variance. The first dimension includes accounting skills, which explain 35%, and the second is planning skills, which explain around 28%. EFA on Organizational Competencies shows that OrgComp is composed of two components that contribute to explaining 62% of the variance. The first dimension is the ability to mobilize financial resources, which explains about 45%, and the second is the ability to mobilize human resources, which in turn explains about 18%. EFA demonstrates that the communicative skills variable is unidimensional. The only dimension retained is interaction and negotiation skills, which explains 61% of the variance. The EFA of the opportunistic competence variable reveals that OppComp is formed by two constitutive dimensions explaining it at 72% of the variance. The first is opportunity recognition, which explains about 50% and the second is creativity, which in turn explains about 21%. Thus, the desirability variable is unidimensional. The items that compose it help explain about 63% of the variance. Furthermore, all constructs have good reliability as their alpha Cronbach's alpha ($\alpha$) is greater than 0.7 according to [76] criteria and [74]. Hence, all measurement scales have good convergent validity as the KMO indices are greater than 0.7 and the values attached to the extracted variances are mostly equal to or higher than the recommended threshold of 0.5. The Bartlett's sphericity test is too significant within valuable chi-squares ($p < 0.05$); thus, it appears that the item correlation matrix is not an identity matrix.

Findings from the discriminant validity (Table 5) showed that the square of the correlation coefficient between the variables was significant at each construct and was lower than the average variance extracted (AVE). The discriminant validity was also verified by adopting the recommendation of [77] that the correlation between measures should not exceed the square root of the AVE. Research by [71] recommend the heterotrait–monotrait ratio (HTMT) of correlations to assess discriminant validity. In such a setting, a HTMT value above 0.90 would suggest that discriminant validity is not present [71]. These findings were in line with [72,74], adhering to a criterion which means that the items used for each

variable in this study measure only what they are supposed to measure, and are completely different from those used to measure other variables.

**Table 4.** Exploratory factor analysis for EP competencies and AD.

| EP competencies | Components | | Communality | Eigen Values | TVD | TVC | αCronbach |
|---|---|---|---|---|---|---|---|
| Strategic Competency. | Accountability | Planning ability | | | | | |
| I know how to read and analyze a balance sheet and draw conclusions | 0.800 | | 0.657 | 2.579 | 35.40 | 63.21 | |
| I can use ratios, indicators, and operating reports to analyze firm performance | 0.754 | | 0.611 | | | | 0.733 |
| I can calculate costs, cost prices, and margins | 0.803 | | 0.673 | | | | |
| I usually work toward specific goals I have set for myself | | 0.867 | 0.752 | 1.214 | 27.81 | | |
| I often visualize the successfully performing of a task before I do it | | 0.620 | 0.537 | | | | |
| I always plan before doing anything | | 0.747 | 0.563 | | | | |
| Psychological competency | Self-efficacy | Motivation to progress | | | | | |
| If someone opposes me, I can find the means and ways to get what I want | 0.842 | | 0.729 | 2.737 | 42.59 | 62.38 | 0.758 |
| I am confident that I could deal efficiently with unexpected events | 0.754 | | 0.605 | | | | |
| I can remain calm when facing difficulties because | 0.737 | | 0.593 | | | | |
| I can usually handle whatever comes my way | 0.711 | | 0.517 | | | | |
| I am willing to be my boss | | 0.772 | 0.612 | 1.007 | 19.79 | | |
| I am motivated to achieve my goals in life | | 0.803 | 0.686 | | | | |
| Organizing competency | Financial Resources | Human Resources | | | | | |
| It is easy for me to find money when needed | 0.762 | | 0.572 | | | | |
| I always join my needs to my income | 0.786 | | 0.681 | 3.578 | 44.72 | | |
| I can identify and meet a firm's financial needs in the short and long term | 0.795 | | 0.578 | | | 62.47 | 0.813 |
| My friends and relatives can easily borrow money from me | 0.772 | | 0.607 | | | | |
| I am financially stable | 0.785 | | 0.621 | | | | |
| It comes to me to delegate tasks | | 0.708 | 0.669 | 1.420 | 17.75 | | |
| I often invite my collaborators to give their opinions to find a solution | | 0.825 | 0.647 | | | | |
| I gather support from my friends anytime it's needed | | 0.742 | 0.622 | | | | |
| Communicative Competence | Interaction and negotiation ability | | | | | | |
| I am very clear when I am presenting ideas verbally | 0.770 | | 0.593 | | | | |
| I always pay attention when someone talks to me | 0.803 | | 0.645 | 2.445 | | 71.73 | 0.788 |
| I am always confident when negotiating to reach an agreement | 0.803 | | 0.644 | | | | |
| I express my opinion when I do not understand the interlocutor | 0.750 | | 0.562 | | | | |
| Opportunistic competency | Opportunity recognition | Creativity | | | | | |
| I am always alert to business opportunities | 0.674 | | 0.756 | | | | |
| I always identify a business opportunity that is not evident to others | 0.796 | | 0.741 | 2.517 | 50.34 | 71.73 | 0.746 |
| I look for information about new ideas on products or services | 0.842 | | 0.601 | | | | |
| It happened to me to write a scientific work | | 0.845 | 0.771 | 1.070 | 21.39 | 0.746 | |
| I have initiated new things that provide satisfaction to people around me | | 0.857 | 0.718 | | | | |
| Agribusiness desirability | Agribusiness desirability (AD) | | | | | | |
| I desire agribusiness activities | 0.715 | | 0.628 | | | | |
| I desire to engage in agribusiness in the future | 0.565 | | 0.573 | | | | |
| Agribusiness is a passion for me | 0.832 | | 0.704 | | | | |
| Agriculture is an acceptable way of life to me | 0.587 | | 0.601 | | | | |
| I can achieve my goals thanks to agribusiness activities | 0.726 | | 0.622 | 5.074 | | 63.43 | 0.738 |
| I wish to venture into agribusiness | 0.692 | | 0.571 | | | | |
| I can develop a successful agricultural business | 0.822 | | 0.753 | | | | |
| Agribusiness is a source of employment for me | 0.762 | | 0.623 | | | | |

TVD: Total variance per dimension. TVC: Total variance for the construct. α Cronbach alpha for the construct. Source: authors' computation.

**Table 5.** Discriminant validity.

| | 1 | 2 | 3 | 4 | 5 | 6 | AVE |
|---|---|---|---|---|---|---|---|
| PSYCOMP (1) | 1 | | | | | | 0.5 |
| STRCOMP (2) | 0.300 ** (0.09) | 1 | | | | | 0.5 |
| ORGCOMP (3) | 0.539 ** (0.29) | 0.363 ** (0.13) | 1 | | | | 0.6 |
| COMCOMP (4) | 0.467 ** (0.22) | 0.276 ** (0.08) | 0.533 ** (0.28) | 1 | | | 0.6 |
| OPPCOMP (5) | 0.164 ** (0.03) | 0.137 ** (0.02) | 0.260 ** (0.07) | 0.195 ** (0.04) | 1 | | 0.5 |
| AD (6) | 0.391 ** (0.15) | 0.213 ** (0.05) | 0.436 ** (0.19) | 0.293 ** (0.09) | 0.166 ** (0.03) | 1 | 0.4 |

Note: **. Correlation is significant at the 0.01 level (2-tailed). Source: authors' computation.

In Table 6, the level of AD among youth is high (67%), with a significant difference between them by gender and living area, which is higher than the median (3). Youth have a significant level of psychological competencies (64%) which implies that they are significantly confident and motivated for success. They are strategically at 57%, which means they have significant knowledge of accounting and planning. Young people are significantly organized (64%), especially in the mobilization of human and financial resources. They are socially competent in learning and adapting (64%). However, they have a low level (51%) and are not significant in creativity and entrepreneurial vigilance. These results imply that young people in South Kivu want to undertake agribusiness given their psychological, strategic, organizational, and communication skills, despite their low rate of creativity, and seize an opportunistic ability to grab new opportunities. The results also demonstrate that youth living in urban areas have slightly (4) higher entrepreneurial competencies than rural youth; namely, psychological, strategic, organizational, and communication skills, which need to be capitalized upon and shared with youth in rural areas. According to [6], rural youth face many constraints related to poverty, and have low access to public services like education and healthcare, which stifles the development of their skills.

**Table 6.** Average values of entrepreneurial potential variables were used.

| Variables | Obs | Min | Max | Male | Female | Urban | Rural | % | (z-Stat) |
|---|---|---|---|---|---|---|---|---|---|
| Agribusiness desirability | 514 | 3.67 | 5 | 3.66 | 3.68 | 3.84 | 3.58 | 66.75 | (23.548) *** |
| Psychological competencies | 514 | 2 | 5 | 3.56 | 3.57 | 3.86 | 3.42 | 64.25 | (23.548) *** |
| Strategic competencies | 514 | 1 | 5 | 3.27 | 3.25 | 3.41 | 3.20 | 56.75 | (10.075) *** |
| Organizing competencies | 514 | 2.25 | 5 | 3.54 | 3.54 | 3.90 | 3.37 | 63.50 | (23.681) *** |
| Communication competencies | 514 | 2 | 5 | 3.59 | 3.56 | 3.90 | 3.42 | 64.50 | (20.044) *** |
| Opportunist competencies | 514 | 1 | 5 | 3.00 | 3.07 | 3.31 | 2.89 | 50.75 | (0.895) |

Note: *** significant at 1%. The figures in parentheses represent the standard deviation. Source: authors' computation.

The results from Table 7 below reveal that psychological skills were positively associated with the AD among young people. The explanation is that the more confident and motivated young people feel, the more they aspire to do agribusiness. Moreover, there is a significant relationship between organizational skills and the AD, implying that the more youth improve their human and financial resource mobilization skills, the more likely they are to engage in agribusiness. Similarly, the AD was significantly explained by opportunistic skills. This result implies that agribusiness is more desired by young opportunists who are both creative and vigilant about their business climate environment. Based on the findings of [78–80], management skills are the set of skills that an entrepreneur in the agricultural sector would use to develop the agricultural business. Furthermore, a previous study [16] showed that psychological capital positively and meaningfully shaped youths' intention to be involved in agricultural entrepreneurship in the eastern part of DR Congo. However, agribusiness desirability was not significantly associated with strategic and communication skills. These results mean that the AD is not a function of the youth's accounting and planning skills; however, the fact that youths have mastered accounting and planning skills is not a necessary and sufficient condition for engaging in agribusiness.

**Table 7.** Results of the model.

|  | Coefficients | Standard Deviation | (T Statistics) |
|---|---|---|---|
| PsyComp -> AD | 0.190 | 0.058 | (3.272) *** |
| StratComp -> AD | 0.010 | 0.049 | (0.194) |
| OrgComp -> AD | 0.285 | 0.063 | (4.519) *** |
| ComComp -> AD | 0.032 | 0.061 | (0.531) |
| OppComp -> AD | 0.207 | 0.049 | (4.215) *** |

Note: *** significant at 1%. The figures in parentheses represent the standard deviation. Source: authors' computation under SmartPLS 3.3.3.

The empirical model is presented in Figure 3 and the important results for the moderation and quality of the structural equation model are in Table 8.

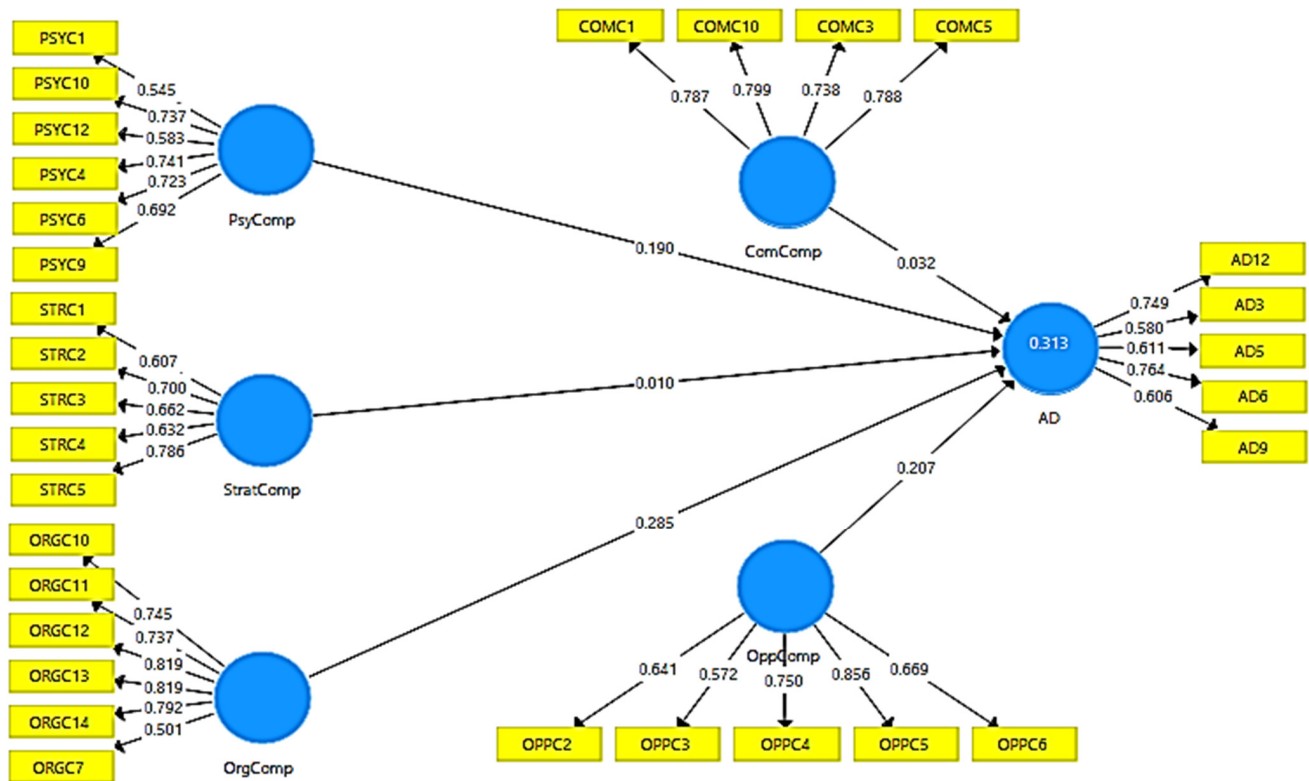

**Figure 3.** The empirical model from PLS SMART.

In addition to the Standardized Root Mean Square Residual (SRMR), chi-square, and NFI values being within acceptable limits [81], Table 8 below indicates that the conceptual model was very good as the empirical model, and (Figure 3) is identical to the saturated model.

**Table 8.** Quality of the structural equation model.

|  | Saturated Model | Estimated Model | Acceptable Limit |
|---|---|---|---|
| SRMR | 0.077 | 0.077 | <0.08 |
| Chi-square | 1763.254 | 1763.254 |  |
| Chi-square/ddl | 3.464 | 3.464 | <5 |
| Normed Fit Index (NFI) | 0.698 | 0.698 | 0 (worse) to 1 (perfect fit) |
| $R^2$ | 0.313 | Adjusted R Square | 0.307 |

Source: authors' computation.

### 3.3. Effect of Socio-Economics Factors on Agribusiness Desirability

Using Equation (1), Table 9 below shows that young male students desire agribusiness more than their peers and that the desirability of agribusiness is significantly but negatively related to the gender of the youth surveyed. Age, marital status, association membership, and youth's monthly income do not affect youth's desire to go into agribusiness. Furthermore, the more the youth is educated, the less likely they willing to go into agribusiness. Youngs who perceive agribusiness as a source of employment are most motivated to engage in agribusiness. In addition, those whose families are involved in farming activities, youth are the most motivated to engage in agribusiness but also when they receive more support from their parents. Similarly, land proportion ownership has a positive and significant effect on youth agribusiness. It should also be noted that the same evidence observed in structural equation analysis of EP on AD was the same with truncated regression within socio-economics and demographics variables as control parameters; psychological, organizational, and opportunistic competencies were positively and significantly correlated to the agriculture entrepreneurship desirability.

**Table 9.** Results from truncated regression between socio-economics and demographic characteristics and EP variables on (AD).

| AD | Coef. | St.Err. | t-value | [95% Conf | Interval] |
|---|---|---|---|---|---|
| GEND (Male) | −0.134 | 0.041 *** | −3.27 | −0.215 | −0.054 |
| AGE | 0.008 | 0.007 | 1.26 | −0.005 | 0.022 |
| MARISTAT | −0.018 | 0.05 | −0.36 | −0.117 | 0.08 |
| EDUC | −0.026 | 0.009 *** | −2.88 | −0.044 | −0.008 |
| ASSOC | 0.049 | 0.037 | 1.33 | −0.023 | 0.122 |
| INCOME | 0 | 0 | 0.41 | 0 | 0 |
| PARSUPPORT | 0 | 0.001 | −0.77 | −0.002 | 0.001 |
| OWNLAND | 0.038 | 0.019 ** | 2.04 | 0.002 | 0.075 |
| PARENTFARM | 0.203 | 0.038 *** | 5.29 | 0.128 | 0.278 |
| AWAREAGRI | 0.178 | 0.05 *** | 3.58 | 0.081 | 0.275 |
| AGRITRAIN | −0.003 | 0.039 | −0.08 | −0.08 | 0.074 |
| AGRISOURCE | 0.232 | 0.069 *** | 3.34 | 0.096 | 0.368 |
| RURALAREA | −0.065 | 0.049 | −1.31 | −0.161 | 0.032 |
| PSYCOMP | 0.19 | 0.043 *** | 4.40 | 0.105 | 0.274 |
| STRCOMP | −0.037 | 0.036 | −1.01 | −0.108 | 0.034 |
| ORGCOMP | 0.269 | 0.054 *** | 5.02 | 0.164 | 0.374 |
| COMCOMP | −0.004 | 0.039 | −0.10 | −0.081 | 0.073 |
| OPPCOMP | 0.058 | 0.027 ** | 2.14 | 0.005 | 0.112 |
| Constant | 0.375 | 0.012 *** | 32.06 | 0.352 | 0.398 |
| Mean dependent var | 3.668 | | SD dependent var | | 0.483 |
| Number of obs | 514.000 | | Chi-square | | 335.408 |
| Prob > chi2 | 0.000 | | Akaike crit. (AIC) | | 490.830 |

Note: *** significant at 1%, ** significant at 5%. The figures in parentheses represent the standard deviation. Source: authors' computation.

## 4. Discussion

Many studies have addressed the entrepreneurial potential issue in the intention to launch SMEs in a rural context, but few authors, especially in the context of South Kivu in the eastern part of DR Congo, have tried to establish the link between entrepreneurial potential competencies combined with most youth's socio-economics, demographics variables, and their agriculture entrepreneurship desirability. Furthermore, this study shows that the level of AD by youth is high, at about 67% despite the trivial effect of strategic and communicative entrepreneurial competencies effect on AD. Hence, as hypothesized from Hypothesis 1 (H1) to hypothesis 6 (H6), most hypotheses (H1, H3, H5, and H6) were verified. Entrepreneurial Potential (EP) of young people in South Kivu was found to have a positive and significant effect on Agribusiness Desirability (AD). Some socio-economics variables, such as gender, education, land ownership, parental involvement in

farming activities, awareness about agriculture initiatives emerging in the community, and perceived agriculture as a source of youth employment, were significant in agribusiness desirability too.

This study reveals that the main psychological predictor components of youth's agribusiness desirability were psychological competency within self-efficacy and motivation to progress. Literally, by focusing on a conceptual model of entrepreneurial competencies and their impacts on rural youth's intention to launch SMEs, the study of [6] attested the same evidence by demonstrating that vigilance, networking ability, individualism, tolerance of ambiguity, and market analysis skills were main predictors of rural youth's intention to launch SMEs. Within the last two decades, the South Kivu province has experienced wars that have disrupted the agricultural system. Besides, credit portfolios, high-interest rates, and guarantees are required for credit applicants. Hence, it is little bit challenging for young people to access loans for entrepreneurial businesses given their poor socioeconomic characteristics, political context, and financial institutions which are sometime limit and do not have often enough credit package for young people to start an agriculture venture and characterized by an awful perception of young people in term of credit management. However, scholars [16,19] attested that youths in South Kivu perceive agribusiness as a socially valued and supported career and are more resilient. Hence, youth with psychological capital: hope, confidence, and optimism, coupled with an important positive supportive role of youth organizing competency on agribusiness desirability within a financial and human resources mobilization ability, are more likely to launch agriculture activities in the DR. Congo context. The results were in line with [40,82–84] who argued that to alleviate uncertainty and ambiguity, and to promote the exchange of ideas and the mobilization of resources needed to create new businesses, policymakers should encourage and facilitate connections between existing and start-up entrepreneurs. The results also revealed that individuals who see good opportunities in the area where they live are more likely to start their own business in the agriculture sector. A policymaker's plan can be designed to enhance the ability to recognize and discover entrepreneurial opportunities in the agriculture sector. According to [41], the lack of opportunistic entrepreneurship competency in seeking information and recognizing market opportunities can be developed and stimulated for significant food security. Additionally, being aware of agricultural initiatives and considering them as an income source was positively linked with AD in the province of South Kivu.

Nowadays, we notice more stakeholders' initiatives (privates, NGOs, and government) in the rural and urban areas, supporting Congolese government efforts in fighting against poverty by setting up agriculture resilience strategies for food security during the pandemic (COVID-19) period to provide more food as border restrictions are increased and young people are playing a significant role. Strong incentives through ICT are therefore taken to restore the middle class through local radio, TV, newspaper, web spot sensitization, and meetings in the communities' associations. This makes youth well aware of what is going on around them and this is an ideal environment that can increase youth agribusiness desirability. It was found that education level is significant but not positive to AD. This has been observed in other studies conducted in Nigeria [80] and in previous studies in South Kivu [21,23]. Furthermore, youth surveyed exhibit different levels of entrepreneurial potential and agribusiness desirability levels according to their gender and living area. This is due to the fact that access to development facilities, like education service, and emancipation activities for young are disproportionate among them [81,82] in rural and urban areas [57]. Furthermore, gender, land ownership, and parental involvement in farming activities as role models were significantly linked with AD. The same results were found by [20,85,86] and are relevant in the context of South Kivu where disincentives are related to market failure, where career orientation is limited or not readily available, and an individual's opinions or perceptions may become the most critical factor in reaching agribusiness decision.

## 5. Theoretical and Methodological Implications

In this study, an innovative conceptual model has been used based on Ajzen's [36] theory of Reasoned Pursuit of Goals and Shapero's notion of a resilient, "self-renewing" economic development for the community [83]. For Shapero, resilience characterizes communities and organizations that are continuously developing. This study states that desire to engage in an agribusiness activity is a function of the perceived individual possibility that the application of entrepreneurial competencies in agribusiness will lead to success and goal attainment. To achieve this, this study used more entrepreneurial active potential competencies based on [6,24,44], using a conceptual model to launch an enterprise in agriculture within gender consideration as well as in rural and urban areas. Specifically, in developing countries within most socio-economics and demographics factors used in the literature as determinant of agriculture entrepreneurship parameters as control variables which is valuable and contributive to the entrepreneurial potential and agriculture entrepreneurship literature as significant portion of recent studies and discussions has focused on Agribusiness Desirability (AD).

## 6. Limitations

Based on the sample of 514 youths in South Kivu, both from rural and urban contexts, the strategic competency and communicative competency of youth can be better harnessed by not only addressing social capital, vigilance, creativity, but forecast ability, as it was the case in this study. Future studies could investigate the roles of other means that foster EP among young people; there is a portion of population within skills that can be built on to strengthen EP. This could further complement the results by investigating the same phenomenon through other theoretical lenses within other demographic parameters such as religiosity, ethnicity, and other control-related independent variables. Furthermore, taking into account in-sampling strategies, people who have been trained, both in theoretical and practical training in agriculture, and other layers of the youth population outside of students, to make more decisions on the appropriate actions for youth sustainable engagement in agribusiness could provide useful insights.

## 7. Conclusions and Recommendations

This study aimed to determine the effect of the level of Entrepreneurial Potential (EP) on the level of Agribusiness Desirability (AD) among youths. The exploratory factor analysis for variable indexes, structural equation, and path analysis were used to determine the correlation between variables. Exploratory analysis reveals that all constructs have good reliability, as their alpha Cronbach's alpha ($\alpha$) was greater than 0.7. All measurement scales have good convergent validity as the KMO indices are greater than 0.7 and the values attached to the extracted variances are mostly equal to or higher than the recommended threshold of 0.5. Thus, six hypotheses were formulated on how entrepreneurial potential competencies and socio-economic and demographic features could affect youth's agribusiness desirability. Results reveal that AD and EP levels among youths were highly significant (67%) and slightly varied according to their gender and living area. Four hypotheses were verified. Among them, psychological, organizational, and opportunistic competencies, in addition to some socio-economic and demographic variables, were positively and significantly correlated with youth agribusiness desirability, except for strategic skills and communication skills. Given these findings, this study recommends the following: (1) stakeholders involved in youth agribusiness initiatives should improve youth entrepreneurial potential by sharing experiences among them, which can lead to improved AD; (2) focus on the use of massive ICT media such as radio, podcasting, and newsletters to create awareness of available agribusiness initiatives emerging in their area, as well as setting up a reasonable youth-friendly land policy, which puts in place capacity-building programs on entrepreneurial and business skills through the development of incubators; and (3) the formalization of youth agribusiness groups that foster capitalizing experiences

between new and accelerated agripreneurial enterprises, with the support of parents and financial institutions, focusing gender sensitivity, in both rural and urban areas.

**Author Contributions:** Conceptualization, G.S.; Supervision, H.S. and P.-M.D.N.; Data curation, G.S.; Formal analysis, G.S.; Funding acquisition, G.S. and Z.B.; Investigation, G.S.; Methodology, G.S., and D.B.; Project administration, Z.B.; Resources, V.M. Validation, H.S., P.-M.D.N. and V.M.; Visualization, G.S. and H.S.; Writing—original draft, G.S. and H.S.; Writing—review and editing, G.S., M.Y., K.B., S.A.M., J.M., D.B., D.-M.A.N., P.-M.D.N., V.M. and H.S. All authors have read and agreed to the published version of the manuscript.

**Funding:** The International Fund for Agriculture Development (IFAD) funded this research under the grant project number PJ 2459, Enhancing Capacity to Apply Research Evidence (CARE) in Policy for Youth Engagement in Agribusiness and Rural Economic Activities in Africa Project in the International Institute of Tropical Agriculture (IITA); 3rd Edition, (2019).

**Institutional Review Board Statement:** The study was conducted according to the guidelines of the International Institute of Tropical Agriculture (IITA) Internal Review Board (IRB).

**Informed consent Statement:** Informed consent was obtained from all participants involved in the study.

**Data Availability Statement:** The data are available upon request.

**Acknowledgments:** We would like to thank the IITA Socio-economic team for their encouragement during all these years of hard work, the IITA team in Uvira station, for according to us a good work environment that allowed this study to get to the end-stage. Thanks also go to the IITA, and especially to IFAD, for funding this study and to IITA- Youth Agripreneurs (IITA-IYA) team in South Kivu for allowing us to be on the field and experience the youth engagement reality in agribusiness. The opinion expressed in this manuscript does not engage authors' affiliations.

**Conflicts of Interest:** The authors declare no conflict of interest.

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
