# Peer review of "Entrepreneurial Potential and Agribusiness Desirability among Youths in South Kivu, Democratic Republic of the Congo"

_sustainability, doi:10.3390/su15010873_

Round 1
Reviewer 1 Report
My decision was to reconsider after a major revision because the size of the sample is good and the authors could have done better work with it. The paper really needs to be revised in several areas, hence I still do not reject it. I outline below the feedback that is also posted in the comments for the authors.
Feedback:
The authors claim in the abstract, a study where they attempt to examine the relationship between Entrepreneurial Potential characteristics and youth desirability to start an enterprise in agriculture. The size of the sample is the most positive aspect of this paper, yet the authors need to undertake major improvements in the paper if they want to be considered for publication.
In the abstract, the present main findings of the study that the population researched exhibit different levels of entrepreneurial potential and agribusiness desirability levels according to their gender and living area. These results seem to bring the research field of entrepreneurship in the context of agribusiness further and it attracts the reader to read the study at hand. Unfortunately, the results of the study only touch on these results in a descriptive manner in Table 6 without any further discussion on these findings. The authors, never bring forward these results to build on the theoretical and practical implications of these findings.
In addition, the present study has several shortcomings in different areas, that need to be reviewed before considering the paper for publicaiton:
Definitions:
The study contributes to the entrepreneurship field and more particularly to entrepreneurship in agriculture. The authors define entrepreneurship in agriculture at a very late stage in the paper. The definition offered (section 2.3) lacks foundation and it implies that entrepreneurship is an activity that is only performed “for-profit”, hence denying other forms of organizations that create value and are part of entrepreneurial manifestations (e.g., non-for-profit social entrepreneurial forms of value creation).
Literature review, Theoretical framework, Conceptual model, and development of hypotheses.
The authors present Section 2 as a literature review when it is a more conceptual background.
In this section, the authors attempt to present and explain the items that form the basis of their conceptual model presented in Figure 1. Nevertheless, the items are further presented in a section entitled theoretical background. The literature reviewed to present the items is deficient.
The section entitled theoretical framework is also deficient. It mainly presents the concepts that constitute the construct of Entrepreneurial Potential. The authors do not elaborate on the theories presented and chosen and as a result, the development of hypotheses is deficient. It just assumes that there will be a positive relationship, but it is not clear where this reasoning comes from.
Theories used
Several theories are mentioned in the study without a clear presentation of the main premises of the main theory used. There is a deficient explanation of why one theory is chosen over the other, or how the study extends on the limitations of certain theories.
It is unclear how the conceptual model to be tested is really rooted in a certain theory. The authors could have graphically depicted the hypothesized relationships among the variables using the premises of the theory chosen for this study. They attempt to depict a conceptual model in Figure 1, which is a confusing representation of the study and not the appropriate way to depict hypotheses about a certain relationship between or among variables.
Gendered comments
The authors explicitly make gendered comments based on gendered assumptions about the role of women in society. The most critical aspect of it, is that they make this comment in the results section, attributing the lack of responses from women due to “…their home tasks in terms of caring for children, housework, and the family in general”. The authors assume that these tasks are for women and not for men. They do not even show evidence on why they draw those conclusions.
The demographic target group presented
The authors classify the youth population as people between 18 and 35 years old while this target group is classified as the adult population as well. It is unclear which source they have used to define their target population.
English grammar issues
Several language-related issues need to be checked.
Level of analysis
The study claims to focus on the individual level of analysis of entrepreneurial potential and to make emphasis the competencies of the individuals, yet the authors make evident throughout the paper that the environment plays an important role in the perception of opportunities and the desirability of founding an agricultural business.
Author Response
Responses to Reviewer I
Open Review
( ) I would not like to sign my review report
(x) I would like to sign my review report
English language and style
( ) Extensive editing of English language and style required
( ) Moderate English changes required
(x) English language and style are fine/minor spell check required
( ) I don't feel qualified to judge about the English language and style
Yes |
Can be improved |
Must be improved |
Not applicable |
|
Is the content succinctly described and contextualized with respect to previous and present theoretical background and empirical research (if applicable) on the topic? |
( ) |
( ) |
(x) |
( ) |
Are all the cited references relevant to the research? |
( ) |
(x) |
( ) |
( ) |
Are the research design, questions, hypotheses and methods clearly stated? |
( ) |
( ) |
(x) |
( ) |
Are the arguments and discussion of findings coherent, balanced and compelling? |
( ) |
( ) |
(x) |
( ) |
For empirical research, are the results clearly presented? |
( ) |
( ) |
(x) |
( ) |
Is the article adequately referenced? |
( ) |
(x) |
( ) |
( ) |
Are the conclusions thoroughly supported by the results presented in the article or referenced in secondary literature? |
( ) |
( ) |
(x) |
( ) |
Comments and Suggestions for Authors
My decision was to reconsider after a major revision because the size of the sample is good and the authors could have done better work with it. The paper really needs to be revised in several areas; hence I still do not reject it. I outline below the feedback that is also posted in the comments for the authors.
Feedback:
The authors claim in the abstract, a study where they attempt to examine the relationship between Entrepreneurial Potential characteristics and youth desirability to start an enterprise in agriculture. The size of the sample is the most positive aspect of this paper, yet the authors need to undertake major improvements in the paper if they want to be considered for publication.
In the abstract, the present main findings of the study that the population researched exhibit different levels of entrepreneurial potential and agribusiness desirability levels according to their gender and living area. These results seem to bring the research field of entrepreneurship in the context of agribusiness further and it attracts the reader to read the study at hand. Unfortunately, the results of the study only touch on these results in a descriptive manner in Table 6 without any further discussion on these findings. The authors, never bring forward these results to build on the theoretical and practical implications of these findings.
Answer: The population researched exhibit different levels of entrepreneurial potential and agribusiness desirability levels according to their gender and living area. The results demonstrate also that youth living in urban areas have slightly entrepreneurial competencies level; psychological, strategic, organizational, and communication skills than rural youth to be capitalize and share with youth in rural area. This is due to the fact that access to development facilities, like education service, emancipation activities for young are disproportionated among them[7,8] in rural and urban areas [9].
In addition, the present study has several shortcomings in different areas, that need to be reviewed before considering the paper for publicaiton:
Definitions:
The study contributes to the entrepreneurship field and more particularly to entrepreneurship in agriculture. The authors define entrepreneurship in agriculture at a very late stage in the paper. The definition offered (section 2.3) lacks foundation and it implies that entrepreneurship is an activity that is only performed “for-profit”, hence denying other forms of organizations that create value and are part of entrepreneurial manifestations (e.g., non-for-profit social entrepreneurial forms of value creation).
Answer: Entrepreneurship is a process of taking a risk by an individual or group of people for creating profit and new values throughout input (work, capital, and land) combination [6]. Furthermore [55,56] define intention as a person's disposition to perform a given behavior such as the individual's desire to pursue a career as an independent. Typically, an individual identifies a need or problem within a community, afterwards becomes a social entrepreneur by starting and developing a social enterprise[55]. Besides, the main motivation for any commercial enterprise is usually to obtain profit; however, the ultimate goal is social impact and plays an important role to bring change within the social sphere. They also assume a mission of creating and perpetuating social as well as economic values [1]. Hence, Entrepreneurship oriented in the agricultural sector becomes congruent to fight against poverty, generate rural employment, reduce migration from rural to urban areas [2] with regards to socio-economic welfare and sustainable agriculture value chain development [3,4].
Literature review, Theoretical framework, Conceptual model, and development of hypotheses.
Theories used
Several theories are mentioned in the study without a clear presentation of the main premises of the main theory used. There is a deficient explanation of why one theory is chosen over the other, or how the study extends on the limitations of certain theories.
Answer: According to the theory of planned behavior (TPB), behavioral intentions, the immediate precursors of behavior, are determined by the subjective norm for behavior and perceived control over behavior [36]. Furthermore, most of the shortcomings of livelihood research and assessment like sustainable livelihoods framework (SLF) are due to the fact that the use of such a framework has been only theoretical [37]. Nevertheless, personality traits and cognitive ability are hidden competencies as argued by [36]. To address the problem with these two kinds of theories, researchers suggest by first understanding what people can do to earn a living; and why they have made these choices as a way to understand how their choices and strategies have been shaped. Researchers have assumed that behavior generally serves as a means to an individual's end. To expand the scope of TPB and increase its explanatory and predictive power, they integrated into a proposed Theory of Reasoned Pursuit of Goals (TRGP). Hence this study opted for the Theory of Reasoned Pursuit of Goals (TRGP) developed by [37], in such a way that agriculture entrepreneurship is a decision of individuals to start a new agriculture business [38]. In addition, the desire to engage in a farming activity depends primarily on the perceived likelihood that the application of entrepreneurial potential skills in the farming activity will lead to achievement and success in agri-food development goals attainment.
It is unclear how the conceptual model to be tested is really rooted in a certain theory. The authors could have graphically depicted the hypothesized relationships among the variables using the premises of the theory chosen for this study. They attempt to depict a conceptual model in Figure 1, which is a confusing representation of the study and not the appropriate way to depict hypotheses about a certain relationship between or among variables.
The authors present Section 2 as a literature review when it is a more conceptual background.
In this section, the authors attempt to present and explain the items that form the basis of their conceptual model presented in Figure 1. Nevertheless, the items are further presented in a section entitled theoretical background. The literature reviewed to present the items is deficient.
The section entitled theoretical framework is also deficient. It mainly presents the concepts that constitute the construct of Entrepreneurial Potential. The authors do not elaborate on the theories presented and chosen and as a result, the development of hypotheses is deficient. It just assumes that there will be a positive relationship, but it is not clear where this reasoning comes from.
Answer: Entrepreneurship is a process of taking a risk by an individual or group of people for creating profit and new values through input (work, capital, and land) combination [6]. Furthermore [55,56] define intention as a person's disposition to perform a given behavior such as the individual's desire to pursue a career as an independent. Typically, an individual identifies a need or problem within a community, and then becomes a social entrepreneur by starting and developing a social enterprise. Besides, the main motivation for any commercial enterprise is usually to obtain profit; however, social impact is the ultimate goal and plays an important role in bringing change to the social sphere, and they assume a mission of creating and perpetuating social as well as economic values [1]. Hence, Entrepreneurship oriented in the agricultural sector becomes congruent to fight against poverty, generate rural employment, reduce rural migration from rural to urban areas [2] with regards to socio-economic welfare and sustainable agriculture value chain development [3,4]. Furthermore, in agriculture entrepreneurship studies, factors were also studied along with demographic factors to understand and profile entrepreneurs better. Hence, agriculture entrepreneurship is not innate; most of the entrepreneurial potential that causes agripreneurs to succeed is acquired through learning, formal and informal manifest entrepreneurial competencies. Hence, researchers [19,36,56] assume that entrepreneurial behavior is planned, reasoned, and controlled in anticipation of likely consequences. The intention to lunch a business, depends upon on the individual's psychological capital [57,58] and entrepreneurial skills [6,33]. The essential ingredients of psychological capital, include self-efficacy, optimism, hope, and resilience [59]. Entrepreneurial skills are; communication, planning, networking, creativity and the ability to identify opportunities in one's environment [6,60]. Resilience is the extent to which individuals are able to bounce back or recover from negative experiences, failures, and adapt to changing and stressful life circumstances [61]. In order to do this, critical skills must be developed, including creativity, flexibility, and adaptability. Hence, [16] research supports a positive relationship between psychological capital and agripreneurial intention. A positive relationship between psychological capital and agripreneurship can be justified by the fact that self-efficacious youth will believe in their abilities to succeed in achieving entrepreneurial behavior [9,25]. In addition, optimistic youth can recognize business opportunities where others see chaos, contradiction, and confusion [32]. Furthermore, the dimension of hope helps them capitalize on these opportunities by setting high goals that they believe they can achieve since they can see a way to success and resilient youth can take risks, bounce back from setbacks and adversity [19,62]. Other researchers have shown that self-efficacy refers to beliefs in one's abilities [63] or the level to which a person feels able to mobilize the motivation, cognitive resources, and courses of action necessary to successfully complete a specific task. By using multivariate analysis approach on assessing farmers' entrepreneurship skills in agriculture and the competitiveness of the small and medium enterprises, [5] and [6] in Iran and [56, 38] in Nigeria categorized five entrepreneurial potential skills which include networking ability, individualism, tolerance of ambiguity, and market analysis were a forecast of rural youth's intention to launch any kind of SMEs. On assessing the effect of norms perceived on agripreneurial intention among youth in Eastern DRC, [13] indicated that psychological capital has affected positively and significantly youths’ agriculture entrepreneurship intention. For [11,14] potential entrepreneurs in the agriculture with higher psychological competency has high risks and drawbacks potentially. Other scholars [2,16,17] demonstrated that entrepreneurial and organizational competencies have positive effect on sustainable development agriculture among farmers. The managerial, technical, and innovative skills of entrepreneurship applied to agriculture have positive results on their yield and well-trained entrepreneurs may become role models to all such disheartened farmers [16]. According to [16], management skills are the complete package of skills that an entrepreneur in the agriculture sector would use to develop the farm business. That means managers perform various agricultural-based activities to mobilize different resources; physical, financial, human resources, and information, to accomplish the agriculture entrepreneurship purpose. [6,40] demonstrated that organizational competencies affect positively enterprise lunching in rural area. For [11], the ability to interact effectively with others has a positive effect on entrepreneurial success. The ability to develop a network between entrepreneurs and other individuals that provide them with resources to create and develop the enterprise has been identified as one of the predictors of entrepreneurial performance [17]. [12,18] attested that in the agriculture sector, individuals who are having entrepreneurs in their networks may also have access to the resources important for starting a business [19]. These resources may include technical knowledge; contact with business class; and emotional support from their community; including friends and family members. All put together, high psychological capital and entrepreneurial skills in agriculture entrepreneurial intention, could allow young people to exhibit high agripreneurial intention. Following the different links presented in framework and literature between agricultural entrepreneurship and the different entrepreneurial competencies, the study hypothesizes that;
Hypothesis 1 (H1): Psychological competency (PSYCOMP) has a positive effect on youth’s agribusiness desirability (AD) among youth in the Eastern part of DR. Congo.
Hypothesis 2 (H2): Strategic competency (STRCOMP) has a positive effect on youth agribusiness desirability among youth in the Eastern part of DR. Congo.
Hypothesis 3 (H3): Organizing competency (ORGCOMP) has a positive effect on youth’s agribusiness desirability among youth in the Eastern part of DR. Congo.
Hypothesis 4 (H4): Communicative competence (COMCOMP) has a positive effect on youth’s agribusiness desirability among youth in the Eastern part of DR. Congo.
Hypothesis 5 (H5): Opportunistic competency (OPPCOMP) has a positive effect on youth’s agribusiness desirability among youth in the Eastern part of DR. Congo.
Figure 1. The author’s conceptual model of entrepreneurial potential and its effect on youth's intention to launch SMEs in rural contexts is inspired by [3]; [4] and [5] and adapted to agriculture entrepreneurship desirability.
Note: GEND: Gender; ASSOC: membership in community association/Group; EDUC: education level; INCOME: Monthly income from a different source of funding; PARSUPORT: rate of parental support in income; OWNLAND: land access; PARENTFARM: parents involved in the farming activity (as a role model); AWAREAGRI: aware of agriculture initiatives emerging in the community; AGRISOURCE: Youth perceived agriculture as a source of employment
Gendered comments
The authors explicitly make gendered comments based on gendered assumptions about the role of women in society. The most critical aspect of it, is that they make this comment in the results section, attributing the lack of responses from women due to “…their home tasks in terms of caring for children, housework, and the family in general”. The authors assume that these tasks are for women and not for men. They do not even show evidence on why they draw those conclusions.
Answer: According to the International Labor Organization, the gender pay gap is as high as 40 percent in rural areas; rural women take on a disproportionate share of unpaid household work, which is neither recognized nor paid. When paid and unpaid work hours are combined, women work much longer hours than men in the household. This affects their emancipation time for other community development activities[6].
The demographic target group presented
The authors classify the youth population as people between 18 and 35 years old while this target group is classified as the adult population as well. It is unclear which source they have used to define their target population.
Answer :For this study, a youth is defined as any individual between the ages of 15 and 35 According to the African Union Charter [1]. However, we start consider youth between the ages of 18 and 35 according to [2], this is the adult age when the individual is able to make a rational choice and is responsible for his or her orientations.
English grammar issues
Several language-related issues checked.
Level of analysis
The study claims to focus on the individual level of analysis of entrepreneurial potential and to make emphasis the competencies of the individuals, yet the authors make evident throughout the paper that the environment plays an important role in the perception of opportunities and the desirability of founding an agricultural business.
Answer: this study, expressed entrepreneurial potential at two stages; individual and country levels. at the individual stage, the main agents in the process of deciding to implement entrepreneurial initiatives, and assume responsibility for the consequences; this perspective is then focused on the cognitions, actions, decisions, aspirations, and emotions of the entrepreneur [40,46]. The second stage; at country level, entrepreneurship is not exclusively the result of an individual’s actions and characteristics, external factors such as the economic, technological, political, and regulatory context play a relevant role. This paper focuses on the individual and consider also that environment factors, could made up entrepreneurial potential competencies [45].
Submission Date
18 September 2022
Date of this review
06 Oct 2022 10:45:19

Reviewer 2 Report
find the file attached

Author Response
All comments addressed was checked, see file in attached.

Reviewer 3 Report
§ The paper an appropriate length.
§ The key messages short, accurate and clear.
§ The text’s meaning is clear.
§ A well-written the introduction.
§ Sets out the argument ....
§ Summarizes recent research related to the topic ...
§ Highlights gaps in current understanding or conflicts in current knowledge ...
§ Establishes the originality of the research aims by demonstrating the need for investigations in the topic area.
§ Original and topicality can only be established in the light of recent authoritative research.
Author Response
Dear authors
The issue of Research is current and very interesting not only for DRC but for many Sothern countries in, congratulations! How can agribusiness attract so many young people who are unemployed and have their families living on small farms in rural places in emerging countries?
However the paper needs a major revision before being considered publishable. Here are some hints to improve the paper.
Regarding the literature review it has been very superfluously compared to the theories mentioned the research attributes to..after in conclusions another theories are mentioned, authors must reorganize and link the new literature review done with the results and thus with the conclusions. Some literature is missing Urquía-Grande & Rubio Alcocer in Agricultural Water Management (2015) or Urquía-Grande et al (2018) in Land Use Policy which treats this problematic applied in ther African countries.
Answer: This study did not abord Agricultural Water Management or Land Use Policy, which deal with this issue in African countries.
Here it was about agribusiness and reasoned pursuit goal
However, the study needs to summarize (erase some variables) so many variables taken into account to influence the agribusiness desirability. The study mixes psychological competences and theories with economic and strategic competences and socio-demographic factors and these make a confusing model where readers can doubt if there are variables with too much noise, complicating the results. Consequently, the hypotheses are too many.
Answer: Theoretical and methodological implications
Answer : In this study, an innovative conceptual model has been used based on Ajzen’s [36] theory of Reasoned Pursuit of Goals and Shapero’s notion of a resilient, "self-renewing" economic development for the community [87]. For Shapero, resilience characterizes communities and organizations that are continuously developing. The study states that, desire to engage in an agribusiness activity is a function of the perceived individual possibility that the application of entrepreneurial competencies in agribusiness will lead to success and goal attainment. To achieve this, the study used more entrepreneurial active potential competencies based on [6,24,43] conceptual model to launch an enterprise in agriculture within gender consideration as well as in rural and urban areas. Specifically in developing countries within most socio-economics and demographics factors used in literature as determinant of agriculture entrepreneurship parameters as control variables which is valuable and contributive to the entrepreneurial potential and agriculture entrepreneurship literature as significant portion of recent studies and discussions has focused on Agribusiness Desirability (AD).
The questionnaire needs to be attached in the appendix for readers to clarify about the questions asked. Or there are only the ones in table 4? Where are the questions defining the AD or the EP? Further, a table is needed with more explanation about the variables description. What questions define a person’s psychological competency (only 6, where are the other 6)? Strategic competencies (5 items)? Opportunistic competencies (6 items, where are the items 1 to 10?)?..etc . Where is the communication competency?
Answer: Supplementary materials statement: The questionnaire is available upon request
Answer: Exploratory Factor Analysis (EFA) was used to discover the factor structure of a measure and to examine its internal reliability. In addition, to improve the model quality, we removed any item whose loading coefficient was lower than 0.5 in the explanation of the construct.
The sample is not justified enough as there are 4 areas with two rural villages where one has 201 respondents..can this be biased?
Answer: This is due to the fact that it is in this area of the province that we find a more significant proportion of universities compared to other areas
Epigraph 3.1. explaining the reliability and validity of the measurement scales must be clarified further.
Answer : Any item that contributed to reducing the overall reliability of the measurement scale was removed. This was assessed by Cronbach's alpha coefficient, which had to be greater than or equal to 0.7. We also deleted any item whose quality of representation (communality) in the measurement scale was lower than 0.5. In addition, to increase the convergent and discriminant validity of the measurement scale, we deleted any item that did not have a structural coefficient of at least 0.5 on one component or that had at least 0.41 on several components at the same time, following [73,74] criteria. Therefore, the study examines the convergence validity of the scales, by assessing the composite reliability (CR) of each measure. Mostly, CR values were greater than 0.7, indicating that the reliability and convergence validity of all scales are confirmed according to the criterion of [74]. The study also calculated validity by measuring the average variance explained (AVE). Thus, the AVE values for the scales ranged around 0.5 and above for all measures.
Table 4 is confusedly organized..needs to be divided in two at least.
|
Components |
Communality |
Eigen Values |
TVD |
TVC |
αCronbach |
|
|
Strategic competency. |
Accountability |
Planning ability |
|
|
|
|
|
|
I know how to read and analyze a balance sheet and draw conclusions |
0.800 |
|
0.657 |
2.579 |
35.40 |
63.21 |
0.733 |
|
I can use ratios, indicators, and operating reports to analyze firm performance |
0.754 |
|
0.611 |
|
|
|||
I can calculate costs, cost prices, and margins |
0.803 |
|
0.673 |
|
|
|||
I usually work toward specific goals I have set for myself |
|
0.867 |
0.752 |
1.214 |
27.81 |
|
|
|
I often visualize the successfully performing of a task before I do it |
|
0.620 |
0.537 |
|
|
|||
I always plan before doing anything |
|
0.747 |
0.563 |
|
|
|
|
|
Psychological competency |
Self-efficacy |
Motivation to progress |
|
|
|
|
|
|
If someone opposes me, I can find the means and ways to get what I want |
0.842 |
|
0.729 |
2.737 |
42.59 |
62.38 |
0.758 |
|
I am confident that I could deal efficiently with unexpected events |
0.754 |
|
0.605 |
|
|
|||
I can remain calm when facing difficulties because |
0.737 |
|
0.593 |
|
|
|||
I can usually handle whatever comes my way |
0 .711 |
|
0.517 |
|
|
|
|
|
I am willing to be my boss |
|
0.772 |
0.612 |
1.007 |
19.79 |
|
|
|
I am motivated to achieve my goals in life |
|
0.803 |
0.686 |
|
|
|||
Organizing competency |
Financial Resources |
Human Resources |
|
|
|
|
|
|
It is easy for me to find money when needed |
0.762 |
|
0.572 |
3.578 |
44.72 |
62.47 |
0.813 |
|
I always join my needs to my income |
0.786 |
|
0 .681 |
|
||||
I can identify and meet a firm’s financial needs in the short and long term |
0.795 |
|
0.578 |
|
||||
My friends and relatives can easily borrow money from me |
0.772 |
|
0.607 |
|
|
|||
I am financially stable |
0.785 |
|
0.621 |
|
|
|
||
It comes to me to delegate tasks |
|
0.708 |
0.669 |
1.420 |
17.75 |
|
||
I often invite my collaborators to give their opinions to find a solution |
|
0.825 |
0.647 |
|
|
|||
I gather support from my friends anytime it's needed |
|
0.742 |
0.622 |
|
|
|
|
|
Communicative Competence |
Interaction and negotiation ability |
|
|
|
|
|
||
I am very clear when I am presenting ideas verbally |
0.770 |
|
0.593 |
2.445 |
|
|
|
|
I always pay attention when someone talks to me |
0.803 |
|
0.645 |
71.73 |
0.788 |
|
||
I am always confident when negotiating to reach an agreement |
0.803 |
|
0.644 |
|
||||
I express my opinion when I do not understand the interlocutor |
0.750 |
|
0.562 |
|
|
|||
Opportunistic competency |
Opportunity recognition |
Creativity |
|
|
|
|
||
I am always alert to business opportunities |
0.674 |
|
0.756 |
2.517 |
50.34 |
71.73 |
0.746 |
|
I always identify a business opportunity that is not evident to others |
0.796 |
|
0.741 |
|
||||
I look for information about new ideas on products or services |
0.842 |
|
0.601 |
|
||||
It happened to me to write a scientific work |
|
0.845 |
0.771 |
1.070 |
21.39 |
0.746 |
|
|
I have initiated new things that provide satisfaction to people around me |
|
0.857 |
0.718 |
|
|
|
||
Agribusiness desirability |
Agribusiness desirability |
|
|
|
|
|
|
|
I desire agribusiness activities |
0.715 |
|
0.628 0.573 0.704 0.601 0.622 0.571 0.753 0.623 |
|
|
|
0.738 |
|
I desire to engage in agribusiness in the future |
0.565 |
|
|
|
63.43 |
|
||
Agribusiness is a passion for me |
0.832 |
|
5.074 |
|
|
|||
Agriculture is an acceptable way of life to me |
0.587 |
|
|
|
||||
I can achieve my goals thanks to agribusiness activities |
0.726 |
|
|
|
||||
I wish to venture into agribusiness |
0.692 |
|
|
|
||||
I can develop a successful agricultural business |
0.822 |
|
|
|
|
|||
Agribusiness is a source of employment for me |
0.762 |
|
|
|
|
|
Findings and its discussion together with the conclusions must be developed further. Findings discussion and conclusions result too simple. There are too many variables, items and hypothesis for the small explanation given. Maybe the model should be run again without the psychological dimension?
Answer
- 5. Discussion
Many studies have addressed the entrepreneurial potential issue in the intention to launch SMEs in a rural context, but few authors especially in the context of South Kivu, the Eastern part of DR Congo have tried to establish the link between entrepreneurial potential competencies combined with most youth’s socio-economics, demographics variables, and their agriculture entrepreneurship desirability. Furthermore, the study shows that the level of AD by youth is high at about 67% despite the trivial effect of strategic and communicative entrepreneurial competencies effect on AD. Hence, as hypothesized from Hypothesis 1 (H1) to hypothesis 6 (H6), most hypotheses (H1, H3, H5, and H6) were verified. Entrepreneurial Potential (EP) of young people in South Kivu, was found to have a positive and significant effect on Agribusiness Desirability (AD). Some socio-economics variables; gender, education, land ownership, parent involvement in farming activities, awareness about agriculture initiatives emerging in the community, and perceived agriculture as a source of youth employment, were significant in agribusiness desirability too.
The study reveals that the main psychological predictor components of youth’s agribusiness desirability were psychological competency within self-efficacy and motivation to progress. Literally, by focusing on a conceptual model of entrepreneurial competencies and their impacts on rural youth’s intention to launch SMEs, the study of [6] attested the same evidence by demonstrating that vigilance, networking ability, individualism, tolerance of ambiguity, and market analysis skills were main predictors of rural youth’s intention to launch SMEs. Within the last two decades, the South Kivu province has experienced wars that have disrupted the agricultural system. Besides, credit portfolios, high-interest rates, and guarantees are required for credit applicants. Hence, it is difficult for young people to access loans for entrepreneurial businesses given their poor socioeconomic characteristics, political context, and financial institutions characterized by a bad perception of young people in term of credit management. However, scholars [16,19] attested that youths in South Kivu perceive agribusiness as a socially valued and supported career and are more resilient. Hence, youth with psychological capital; hope, confidence, and optimism coupled with an important positive supportive role of youth organizing competency on agribusiness desirability within financial and human resources mobilization ability are more likely to launch agriculture activities in the DR. Congo context. The results were in line with [83] and [40]. Who argued that to alleviate uncertainty and ambiguity, and to promote the exchange of ideas and the mobilization of resources needed to create new businesses, policymakers should encourage and facilitate connections between existing and start-up entrepreneurs. The results revealed also that individuals who see good opportunities in the area where they live are more likely to start their own business in the agriculture sector. A policymaker’s plan can be designed to enhance the ability to recognize and discover entrepreneurial opportunities in the agriculture sector. According to [41], the lack of opportunistic entrepreneurship competency in seeking information and recognizing market opportunities can be developed and stimulated for significant food security. Additionally, being aware of agricultural initiatives and considering it as an income source was positively linked with AD in South Kivu province.
Nowadays, we notice more stakeholders’ initiatives (privates, NGOs, and Government) in the rural and urban areas, supporting Congolese government efforts in fighting against poverty by setting up agriculture resilience strategies for food security during the pandemic (Covid-9) period to provide more food as border restriction are increased and young people are playing a significant role. Strong incentives through ICT; used media are therefore taken to restore the middle class through local radio, TV, newspaper, web spot sensitization, and meetings in the communities’ associations. This makes youth well aware of what is going on around them and this is an ideal environment that can increase youth agribusiness desirability. It was found that education level is significant but not positive to AD. This has been observed in other studies conducted in Nigeria [84] and in previous studies in South Kivu [21,23]. Furthermore, youth surveyed exhibit different levels of entrepreneurial potential and agribusiness desirability levels according to their gender and living area. This is due to the fact that access to development facilities, like education service, emancipation activities for young are disproportionated among them[85,86] in rural and urban areas [61] . Furthermore, gender, land ownership, and parent involvement in farming activities as role models were significantly explained AD. The same results were found by [20] and are relevant in the context of South Kivu where disincentives are related to market failure, where career orientation is limited or not readily available, and an individual's opinions or perceptions may become the most critical factor in reaching agribusiness decision.
- Limitations
Based on the sample of 514 youths in South Kivu, both from rural and urban contexts, the strategic competency and communicative competency of youth can be better harnessed by not only addressing social capital, vigilance, creativity, and forecast ability as it was the case in this study. Future studies could investigate the roles of other means that foster EP among young people; there is a portion of population within skills that can be built on to strengthen EP. This could further complement the results by investigating the same phenomenon through other theoretical lenses within other demographic parameters such as religiosity, ethnicity, and other control-related independent variables. Furthermore, taking into account in sampling strategies, people who have been trained, both theoretical and practical training in agriculture, and other layers of the youth population despite students, to make more decisions on the appropriate actions for youth sustainable engagement in agribusiness could provide useful insights.
- Conclusions and Recommendations
This study aimed to determine the effect of the level of Entrepreneurial Potential (EP) on the level of Agribusiness Desirability (AD) among youths. The exploratory factor analysis for variable indexes, structural equation, and path analysis were used to determine the correlation between variables. Exploratory analysis reveals that all constructs have good reliability as their alpha Cronbach's alpha (α) was greater than 0.7. All measurement scales have good convergent validity as the KMO indices are greater than 0.7 and the values attached to the extracted variances are mostly equal to or higher than the recommended threshold of 0.5. Thus, 6 Hypotheses were formulated on how entrepreneurial potential competencies and socio-economics and demographic features could affect youth’s agribusiness desirability. Results reveal that AD and EP levels among youths were high (67%) significant and slightly varied according to their gender and living area. 4 hypotheses were verified. Among them, psychological, organizational, and opportunistic competencies and some socio-economic and demographic variables were positively and significantly correlated with young agribusiness desirability. Except for strategic skills and communication skills. Given these findings, the study recommended stakeholders involved in youth agribusiness initiatives improve youth entrepreneurial potential by sharing experiences among them. This can lead to improved AD. Focus on massive ICT media used; radio, podcasting, and newsletters to create awareness of available agribusiness initiatives emerging in their area as well as setting up a reasonable youth-friendly land policy, put in place capacity-building programs on entrepreneurial and business skills through the development of incubators. Formalized youth agribusiness groups that foster capitalizing experiences between new and accelerated agripreneurs enterprises from within parent and financial institutions' support within gender sensitivity as well in rural and urban areas.
Some proof editing is needed as there are typos along all the text..when explain the survey there’s a “epuration”¿?
Answer: Checked, see file in attached
Page numbering is also needed
Answer: Checked, see file in attached
Therefore a major revision is needed. Wish you luck in the hard work required.
Answer: Major revision Checked, see file in attached
